# Divergolides T–W with Apoptosis-Inducing Activity from the Mangrove-Derived Actinomycete *Streptomyces* sp. KFD18

**DOI:** 10.3390/md17040219

**Published:** 2019-04-11

**Authors:** Li-Man Zhou, Fan-Dong Kong, Qing-Yi Xie, Qing-Yun Ma, Zhong Hu, You-Xing Zhao, Du-Qiang Luo

**Affiliations:** 1College of Life Science, Key Laboratory of Medicinal Chemistry and Molecular Diagnosis of Ministry of Education, Hebei University, Baoding 071002, China; zhouliman88@126.com; 2Hainan Key Laboratory for Research and Development of Natural Product from Li Folk Medicine, Institute of Tropical Bioscience and Biotechnology, Chinese Academy of Tropical Agricultural Sciences, Haikou 571101, China; kongfandong@itbb.org.cn (F.-D.K.); xieqingyi@itbb.org.cn (Q.-Y.X.); maqingyun@itbb.org.cn (Q.-Y.M.); 3Guangdong Provincial Key Laboratory of Marine Biotechnology, Department of Biology, Shantou University, Shantou 515063, China; hzh@stu.edu.cn

**Keywords:** mangrove-derived actinomycete, ansamycins, divergolides, apoptosis-inducing activity

## Abstract

Four new ansamycins, named divergolides T–W (**1**–**4**), along with two known analogs were isolated from the fermentation broth of the mangrove-derived actinomycete *Streptomyces* sp. KFD18. The structures of the compounds, including the absolute configurations of their stereogenic carbons, were determined by spectroscopic data and single-crystal X-ray diffraction analysis. Compounds **1**–**4** showed cytotoxic activity against the human gastric cancer cell line SGC-7901, the human leukemic cell line K562, the HeLa cell line, and the human lung carcinoma cell line A549, with **1** being the most active while compounds **5** and **6** were inactive against all the tested cell lines. Compounds **1** and **3** showed very potent and specific cytotoxic activities (IC_50_ 2.8 and 4.7 µM, respectively) against the SGC-7901 cells. Further, the apoptosis-inducing effect of **1** and **3** against SGC-7901 cells was demonstrated by two kinds of staining methods for the first time.

## 1. Introduction

Ansamycins are a class of bioactive macrolides that have been isolated from actinomycetes [1,2,3,4]. The most representatives of them are geldanamycin with HSP90 inhibitory activity [1], rifamycin with antibacterial activity [2], and maytansinoid with anticancer activity [3]. Divergolides represent a family of ansamycins with a 19-membered naphthalenic ansamacrolactam skeleton, which was first discovered from *Streptomyces* sp. HKI0576 and reported in 2011. Until now, a total of 19 members (divergolides A–S) of this family has been reported [5,6,7]. Many divergolides have shown cytotoxic and antibacterial activities [5,6,7,8].

As part our ongoing search for new bioactive secondary metabolites from marine microorganisms [9,10,11,12], *Streptomyces* sp. KFD18 attracted our attention for its ability to produce a series of metabolites with UV absorption bands around 275 and 305 nm, detected by HPLC analysis. Subsequent chemical investigations on the EtOAc extract from the fermentation broth of this strain led to the isolation and identification of four new ansamycins, named divergolides T–W (**1**–**4**), as well as two known analogues 6,7-*epi*-24,25-dihydro-divergolide U (**5**) [8] and divergolide E (**6**) [7] (Figure 1). Herein, the structures and bioactivities of these compounds are reported.

## 2. Results and Discussion

Compound **1** was obtained as a yellow crystal, and was found to have the molecular formula C_31_H_37_NO_7_ from the HRESIMS *m/z* 536.2641 [M + H]^+^. The UV spectrum showed characteristic absorption bands around 221 and 240 nm. The IR absorptions at 3414 and 1663 cm^−1^ revealed the presence of a hydroxy and carbonyl group, respectively. The ^1^H and ^13^C NMR spectra (Appendix A) along with the HSQC spectra (Appendix A) revealed the presence of five methyls, five sp^3^ methylenes, nine methines (including five sp^2^ and one oxygenated sp^3^), twelve non-protonated carbons (including two ketone carbonyls, two ester or amide carbonyls, seven aromatic or olefinic carbons, and one hydroxylated carbon). Comparison of the above data with those of the known analogue **5** [8] suggested that their planar structures were quite similar, except that the hydroxy at C-7 was absent, and the ∆^24^ double bond of **5** was hydrogenated in **1**. In the ^1^H-^1^H COSY spectrum (Figure 2) of **1**, correlations of H-26/H-25/H-27 and H-25/H-24/H-6/H-7 were observed, which further confirmed the above deduction. The remaining substructure of **1** was found to be identical to that of **5** by analysis of the 2D NMR data.

The large *J* value (15.6 Hz) of H-8/H-9 (Table 1) suggested the *E* configuration of the ∆^8^ double bond, while the relative downfield shift (*δ*_C/H_ 21.4/2.17) of the allylic methyl C-4a [13] and ROESY cross-peak (Figure 3) between H-4a and H-3 (*δ*_H_ 6.60) suggested the *Z* configuration of the ∆^3^ double bond. Additionally, in the ROESY spectrum (Figure 3), correlations of H-10/H-8/H-24/H-2 and H-9/H-10a led to the assignment of the full relative configuration of compound **1,** as shown in Figure 3. To support the above assignment and determine the absolute configuration of **1**, a single-crystal X-ray diffraction pattern was obtained using the anomalous scattering of Cu Kα radiation (Figure 4), allowing an explicit assignment of the absolute structure as 2*R*, 6*S*, 10*R*, and 19*R* based on the Flack parameter of −0.05(8).

Compound **2** was determined to have a molecular formula of C_31_H_37_NO_8_ based on HRESIMS data, with one oxygen atom more than that of **1**. The UV spectrum of **2** was nearly identical to that of **1**, suggesting that **2** was a homologue of **1**. Their NMR data (Table 1 and Table 2) were also quite similar, except for the replacement of CH_2_-7 signals in **1** by signals for a hydroxylated sp^3^ methine (*δ*_C/H_ 70.5/3.90) in **2**. In the COSY spectrum (Appendix A), correlations of this hydroxylated sp^3^ methine with H-8 (*δ*_H_ 4.06) and H-6 (*δ*_H_ 4.99) were observed, further confirming that CH_2_-7 in **1** was oxidized to a hydroxylated methine in **2**. The similar *J* values (Table 1) and ROESY data (Figure 3) between **1** and **2** suggested that both compounds had the same configuration at the stereogenic centers C-2, C-6, C-10, and C-19 and double bonds ∆^3^ and ∆^8^. The syn orientation between H-6 and H-7 was deduced from their small vicinal coupling constant (*J* = 2.6 Hz) [12].

Compounds **3** and **4** had the same molecular formula of C_31_H_37_NO_7_ as that of **1**. The ^1^H and ^13^C NMR data (Appendix A) of **3** and **4** were also quite similar to those of **1**. Detailed analysis of the ^1^H-^1^H COSY and HMBC data (Appendix A) of **3** and **4** revealed the same H/H and H/C correlational relationship as those of **1**, indicating that **3** and **4** shared the same planar structure with **1**. However, unlike the ROESY data of **1** and **2**, the absence of correlations (Appendix A) between H-2 and H-24 (*δ*_H_ 1.15 and 1.20, respectively) in **3** and **4** revealed the H-2 protons had opposite orientations as compared to those of **1** and **2**. The syn orientation of H-2 and OH-19 in **3** and **4** was deduced by comparison of the NMR data with those of hygrocins D and F [13]. The above assignment was further supported by the phenomenon that H-2 signals (*δ*_H_ 6.36 and 5.89, respectively) of **3** and **4** resonated upfield [13] compared to those (*δ*_H_ 6.60 and 6.67, respectively) of **1** and **2**. Further, in the ROESY spectra (Figure 3), correlations of H-4a/H-3 of **3** while H-4a/H-2 of **4** were observed, revealing the *Z* and *E* configuration of ∆^3^ double bond in **3** and **4**, respectively.

Compounds **1**–**6** were tested for their cytotoxic activity against the human gastric cancer cell line SGC-7901, the human leukemic cell line K562, the HeLa cell line, and the human lung carcinoma cell line A549. The results (Table 3) showed that compounds **1**–**4** exhibited cytotoxic activity against SGC-7901 (IC_50_ = 2.8, 9.8, 4.7, and 20.9 μM, respectively), K562 (IC_50_ = 6.6, 9.0, 7.6, and 16.3 μM, respectively), HeLa (IC_50_ = 9.6, >50, 14.1, and 29.5 μM, respectively), and A549 (IC_50_ = 14.9, 24.7, 20.9, and 33.2 μM, respectively) cell lines, with **1** being the most active while compounds **5** and **6** were inactive against all the tested cell lines. The above data showed that hydroxylation at C-7 or inversion of the configuration at C-2 or ∆^3^ double bond in compound **1** could significantly reduce cytotoxic activity.

In order to determine whether the compounds could induce apoptosis, we used two kinds of staining methods. Double staining with acridine orange-ethidium bromide (AOEB) allows for differentiation of live, apoptotic, and necrotic cells [14]; live cells have green, regular-sized nuclei. Green or yellow-green nuclear condensation or fragmentation identifies early apoptotic cells, and orange or red staining identifies late apoptotic or necrotic cells. DAPI staining can reveal the typical apoptotic feature: a condensed nucleus and apoptotic body formation [15]. After SGC-7901 cells were cultured with compounds **1** and **3** at double the IC_50_ concentration for 48 h. AOEB staining showed us that the cells were dyed yellow-green or orange. DAPI staining showed that many cells had typical apoptotic features (Figure 5). All staining results indicated that compounds **1** and **3** had apoptosis-inducing activity against SGC-7901. The apoptosis-inducing activity of divergolides has been reported for the first time.

## 3. Experimental Section

### 3.1. General Experimental Procedure

Optical rotations were measured with a JASCO P-1020 digital polarimeter. The IR spectra were obtained with a Nicolet Nexus 470 spectrophotometer as KBr discs. The UV spectra were obtained with a Beckman DU 640 spectrophotometer. The NMR spectra were recorded on a Bruker AV-500 spectrometer, with a CD_3_OD solvent peak signal as the chemical shift reference. All compounds isolated underwent NMR analysis using about 500 μL CD_3_OD solvent. HREIMS data were acquired on a Micromass Autospec-Ultima-TOF, API QSTAR Pulsar 1, or Waters Autospec Premier spectrometer. Semi-preparative HPLC separation used octadecyl silane (ODS) columns (YMC-pack ODS-A, 10 × 250 mm, 5 μm, 4 mL/min) for separation. Thin-layer chromatography (TLC) and column chromatography (CC) were carried out on precoated silica gel GF_254_ (10–40 μm, Qingdao Marine Chemical Inc., Qingdao, China) and silica gel (200–300 mesh, Qingdao Marine Chemical Inc., Qingdao, China), respectively.

### 3.2. Strain and Fermentation

The strain *Streptomyces* sp. KFD18 was isolated from Mangrove sediment, collected from Danzhou, Hainan province, in China, which was identified based on the 16S rRNA gene sequences (GenBank accession No. MK478900, Appendix A) of the single colonies. A reference culture of *Streptomyces* sp. KFD18 was deposited in our laboratory and was maintained at −80 °C. *Streptomyces* sp. KFD18 was cultured in seawater medium containing 1% starch, 0.1% peptone, and 0.2% CaCO_3_ on a rotary shaker (180 rpm) at 28 °C for 4 d to afford a seed culture. Fermentation (30 L) was performed using the same medium on a rotary shaker (180 rpm) at 28 °C for 10 d.

### 3.3. Extraction and Isolation

The fermented cultures were extracted with three-fold volumes of EtOAc, then the EtOAc solutions were combined and evaporated under reduced pressure to produce a dark brown, solid, crude extract (2.9 g). The extract was fractionated by a silica gel VLC column using different solvents of increasing polarity, from MeOH/H_2_O (1:4) to MeOH/H_2_O (1:0), to yield seven fractions (Frs. 1−7). Fr. 5 (87 mg) was subjected to semipreparative HPLC (YMC-pack ODS-A, 5 μm; 10 × 250 mm; 50% MeCN/H_2_O; containing 0.1% TFA; 4 mL/min) to afford compounds **1** (*t*_R_ 19.4 min; 14.2 mg) and **4** (*t*_R_ 23.4 min; 4.3 mg). Fr. 6 (264 mg) was subjected to semipreparative HPLC (YMC-pack ODS-A, 5 μm; 10 × 250 mm; 70% MeCN/H_2_O; containing 0.1% TFA; 4 mL/min) to afford compound **3** (*t*_R_ 13.1 min; 6.4 mg). Fr. 4 (124 mg) was purified by semipreparative HPLC (YMC-pack ODS-A, 5 μm; 10 × 250 mm; 40% MeCN/H_2_O; containing 0.1% TFA; 4 mL/min) to afford compound **2** (*t*_R_ 9.6 min; 3.1 mg) and compound **5** (*t*_R_ 11.3 min; 7.7 mg). Fr. 3 (214 mg) was purified by Sephadex LH-20 chromatography and eluted with MeOH to afford compound **6** (13.8 mg)

Divergolide T (**1**): Colorless crystal; [*α*${]}_{D}^{25}$ −190 (*c* 0.1, MeOH); UV (MeOH) *λ*_max_ (log *ε*): 305.0 (3.70), 275.0 (3.68) nm; IR (KBr) *ν*_max_ (cm^−1^): 3414, 2957, 2855, 1663, 1573, 1194, and 1144. ^1^H and ^13^C NMR data, Table 1 and Table 2; HRESIMS *m/z* 536.2641 [M + H]^+^ (calculated for C_31_H_38_O_7_N, 536.2643).

Divergolide U (**2**): White powder; [*α*${]}_{D}^{25}$ +60 (*c* 0.1, MeOH); UV (MeOH) *λ*_max_ (log *ε*): 305.0 (3.72), 275.0 (3.69) nm; IR (KBr) *ν*_max_ (cm^−1^): 3444, 2925, 2855, 1677, 1442, 1199, and 1141. ^1^H and ^13^C NMR data, Table 1 and Table 2; HRESIMS *m/z* 550.2438 [M − H]^+^ (calculated for C_31_H_36_O_8_N, 550.2446).

Divergolide V (**3**): White powders; [*α*${]}_{D}^{25}$ +118 (*c* 0.1, MeOH); UV (MeOH) *λ*_max_ (log *ε*): 305.0 (3.75), 275.0 (3.70) nm; IR (KBr) *ν*_max_ (cm^−1^): 3413, 2926, 1649, 1583, 1334, 1243, 1146, and 1058. ^1^H and ^13^C NMR data, Table 1 and Table 2; HRESIMS *m/z* 536.2640 [M + H]^+^ (calculated for C_31_H_38_O_7_N, 536.2643).

Divergolide W (**4**): White powders; [*α*${]}_{D}^{25}$ +72 (*c* 0.1, MeOH); UV (MeOH) *λ*_max_ (log *ε*): 305.0 (3.68), 275.0 (3.66) nm; IR (KBr) *ν*_max_ (cm^−1^): 3442, 2926, 2961, 1673, 1577, 1199, and 1138. ^1^H and ^13^C NMR data, Table 1 and Table 2; HRESIMS *m/z* 534.2490 [M − H]^−^ (calculated for C_31_H_36_O_7_N, 534.2497).

X-ray Crystal Data for 1: Colorless crystals of **1** were obtained in the mixed solvent of MeOH. Crystal data of **1** were obtained on a Bruker D8 QUEST diffractometer (Bruker) with graphite monochromated Cu Kα radiation (*λ* = 1.54178 Å). Crystallographic data for **1** were deposited in the Cambridge Crystallographic Data Center as supplementary publication number CCDC 1893418. These data can be obtained free of charge from The Cambridge Crystallographic Data Centre via www.ccdc.cam.ac.uk/data_request/cif.

Crystal data for **1**. Monoclinic, C_31_H_37_NO_7_; space group P 1 21 1 with *a* = 12.5723(5) Å, *b* = 14.6723(6) Å, *c* = 17.4900(8) Å, *V* = 3226.3(2) Å^3^, *Z* = 1, *D*_calcd_ = 1.109 g/cm^3^, *μ* = 0.691 mm^−1^, and *F*(000) = 1077. *T* = 296.15 K. *R*1 = 0.0526 (*I* > 2σ(I)), *wR*2 = 0.1480 (all data), *S* = 1.021. Absolute structure parameter: −0.05(8). The structures were solved using ShelXS. The structural solutions were found by direct methods and refined using the ShelXL package by least squares minimization. The final structures were examined using the Addsym subroutine of PLATON to assure that no additional symmetry could be applied to the models. All non-hydrogen atoms were refined with anisotropic thermal factors.

### 3.4. Bioassays for Cytotoxic and Apoptosis-Inducing Activity

The cytotoxic activities of compounds **1**–**6** were tested in vitro by using the MTT method optimized by Chuan et al. [16]. Imatinib and adriamycin were used as the positive controls, and a medium with 4‰ DMSO was used as the negative control in the bioassay test. For AOEB staining, SGC-7901 cells were cultured in 96-well cell culture plates. After 48 h incubation, the culture medium was removed and washed with PBS three times. AO and EB were added to a final concentration of 2 μg/mL each. For DAPI staining, cells were fixed with 4% paraformaldehyde solution for 10 min, incubated with 0.1% TritonX-100 on ice for 30 min, and then washed with PBS three times. DAPI was added to a final concentration of 1 μg/mL each. The pictures were taken using a fluorescence microscope.

## 4. Conclusions

In conclusion, four new ansamycins (**1**–**4**) and two known analogs (**5** and **6**) were isolated from the fermentation broth of mangrove-derived actinomycete *Streptomyces* sp. KFD18. Compounds **1**–**4** exhibited cytotoxic activity against SGC-7901(IC_50_ = 2.8, 9.8, 4.7, and 20.9 μM, respectively), K562 (IC_50_ = 6.6, 9.0, 7.6, and 16.3 μM, respectively), HeLa (IC_50_ = 9.6, >50, 14.1, and 29.5 μM, respectively), and A549 (IC_50_ = 14.9, 24.7, 20.9, and 33.2 μM, respectively) cell lines, with **1** being the most active while compounds **5** and **6** were inactive against all the tested cell lines. The two most active compounds, **1** and **3,** could induce apoptosis of SGC-7901 cells.

## Figures and Tables

**Figure 1 marinedrugs-17-00219-f001:**
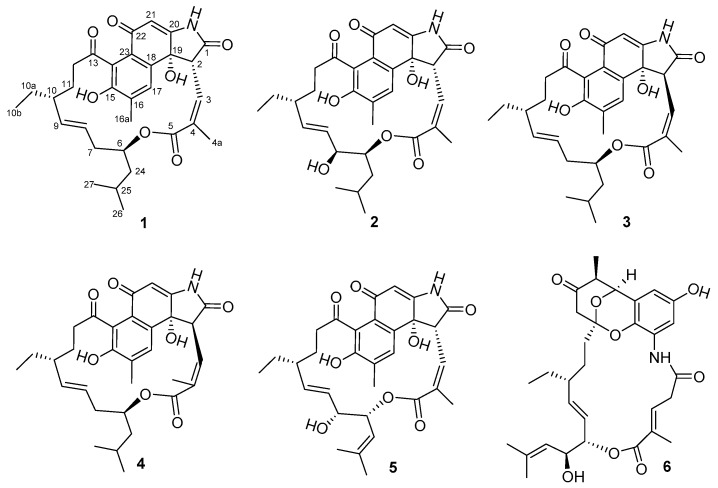
Structures of compounds **1**–**6**.

**Figure 2 marinedrugs-17-00219-f002:**
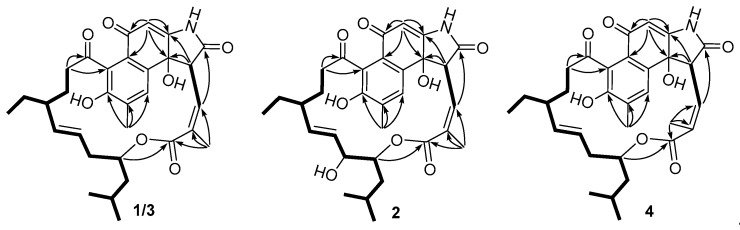
Key COSY (

) and HMBC (→) correlations of **1**–**4**.

**Figure 3 marinedrugs-17-00219-f003:**
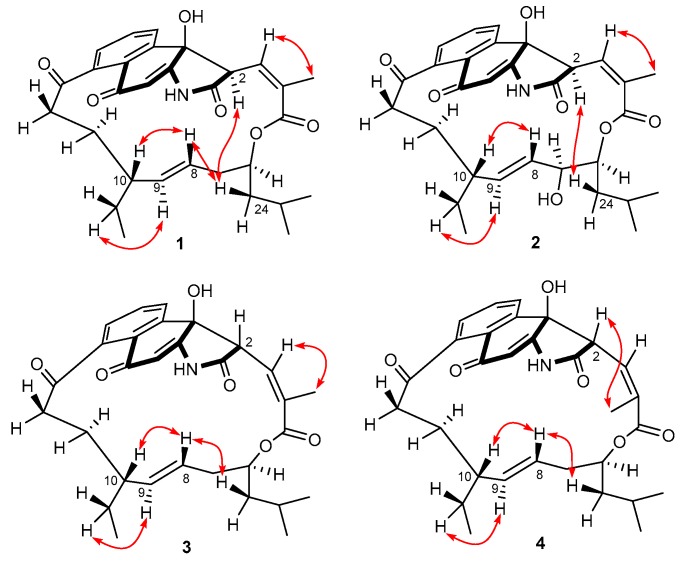
Key ROESY correlations of **1**–**4**.

**Figure 4 marinedrugs-17-00219-f004:**
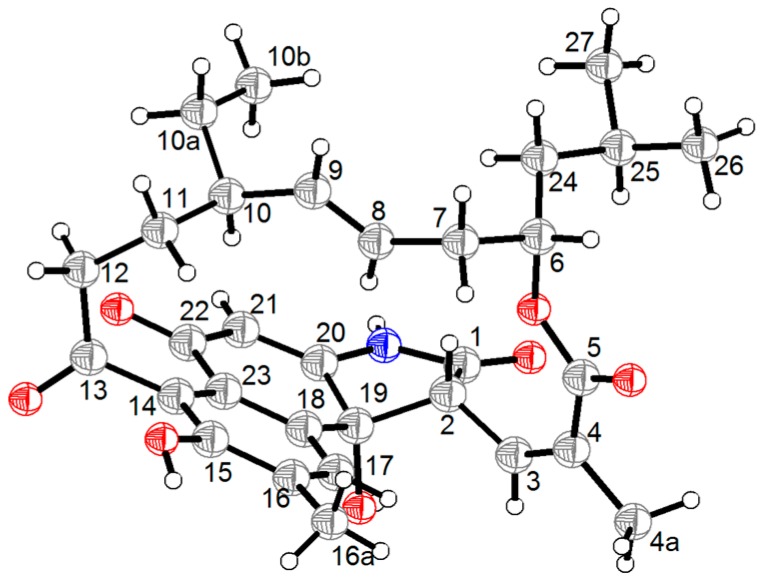
ORTEP diagram of **1**.

**Figure 5 marinedrugs-17-00219-f005:**
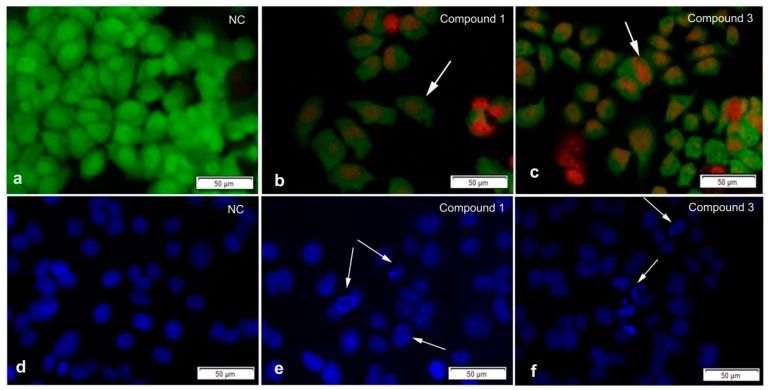
The staining results of compounds **1** and **3** on SGC-7901. Acridine orange-ethidium bromide (AOEB) staining on SGC-7901 cells at 48 h after compound addition (**a**–**c**). DAPI staining on SGC-7901 cells at 48 h after compound addition (**d**–**f**). The concentrations of compounds **1** and **3** were 5.6 μM and 9.4 μM, respectively. NC: Negative control, DMSO of the same volume.

**Table 1 marinedrugs-17-00219-t001:** ^13^C NMR data for **1**–**4** in CD_3_OD.

Position	1	2	3	4
	*δ* _C_	*δ* _C_	*δ* _C_	*δ* _C_
1	177.2, C	177.1, C	177.2, C	176.8, C
2	55.2, CH	55.3, CH	55.8, CH	55.3, CH
3	131.7, CH	132.9, CH	126.7, CH	132.5 CH
4	136.8, C	136.1, C	138.4, C	135.0, C
4a	22.0, CH_3_	22.1, CH_3_	21.4, CH_3_	13.5, CH_3_
5	168.0, C	167.7, C	169.5, C	167.9, C
6	74.6, CH	76.7, CH	74.5, CH	74.0, CH
7	36.1, CH_2_	70.5, CH	36.6, CH_2_	36.3, CH_2_
8	125.1, CH	128.2, CH	126.3, CH	125.3, CH
9	139.6, CH	136.1, CH	138.8, CH	138.6, CH
10	46.0, CH	45.9, CH	44.2, CH	43.2, CH
10a	26.9, CH_2_	27.0, CH_2_	29.3, CH_2_	25.6, CH_2_
10b	13.2, CH_3_	13.2, CH_3_	12.7, CH_3_	11.1, CH_3_
11	31.7, CH_2_	31.5, CH_2_	34.8, CH_2_	31.6, CH_2_
12	40.5, CH_2_	40.5, CH_2_	42.5, CH_2_	42.0, CH_2_
13	212.4, C	212.4, C	212.1, C	212.1, C
14	130.1, C	130.2, C	127.8, C	130.6, C
15	153.5, C	153.5, C	153.4, C	152.9, C
16	132.6, C	132.9, C	133.1, C	133.4, C
16a	17.0, CH_3_	17.0, CH_3_	17.0, CH_3_	16.9, CH_3_
17	130.8, CH	130.8, CH	132.8, CH	131.9, CH
18	134.3, C	136.1, C	134.3, C	135.0, C
19	73.7, C	73.8, C	73.6, C	75.2, C
20	164.8, C	164.7, C	164.6, C	164.6, C
21	103.9, CH	104.0, CH	103.4, CH	104.3, CH
22	185.4, C	185.4, C	185.7, C	185.7, C
23	129.8, C	129.9, C	130.4, C	130.6, C
24	42.6, CH_2_	38.4, CH_2_	41.5, CH_2_	41.9, CH_2_
25	25.5, CH	25.7, CH	25.3, CH	25.6, CH
26	22.7, CH_3_	22.2, CH_3_	22.1, CH_3_	22.4, CH_3_
27	23.0, CH_3_	23.8, CH_3_	23.7, CH_3_	23.6, CH_3_

**Table 2 marinedrugs-17-00219-t002:** ^1^H NMR data for **1**–**4** in CD_3_OD.

Position	1	2	3	4
	*δ*_H_ (*J* in Hz)	*δ*_H_ (*J* in Hz)	*δ*_H_ (*J* in Hz)	*δ*_H_ (*J* in Hz)
2	4.74, d (10.9)	4.84, d (10.6)	4.09, d (10.9)	4.06, d (8.4)
3	6.60, dq (10.9, 1.6)	6.67, dq (10.6, 1.6)	6.36, dq (10.8, 1.6)	5.89, dq (8.4, 1.6)
4a	2.20, d (1.6)	2.21, d (1.6)	2.17, d (1.6)	2.08, d (1.0)
6	5.05, m	4.99, m	5.04, m	4.87, m
7	1.96, m	3.90, ddd (2.74, 2.6, 2.6)	2.15, m	2.25, m
	2.15, m		2.15, m	2.14, m
8	3.93, ddd (15.3, 10.2, 3.6)	4.06, dd (15.6, 2.8)	3.78, ddd (15.6, 6.0, 6.0)	4.77, ddd (15.6, 9.1, 4.9)
9	5.01, dd (15.3, 9.3)	5.24, dd (15.6, 9.3)	4.87, dd (15.6, 9.4)	5.24, dd (15.6, 7.7)
10	1.32, overlap	1.37, overlap	1.46, overlap	1.78, m
10a	0.89, m	0.92, m	1.02, m	1.43, overlap
	0.89, m	1.49, overlap	1.34, overlap	1.18, overlap
10b	0.66, t (7.4)	0.65, t (7.4)	0.73, t (7.4)	0.77, t (7.5)
11	1.35, m	1.37, overlap	1.68, m	1.29, overlap
	1.46, overlap	1.49, overlap	1.25, m	1.57, m
12	2.61, m	2.62, m	2.64, ddd (14.0, 11.3, 2.8)	2.46, m
	2.90, m	2.99, m	2.46, ddd (14.0, 7.4, 2.9)	2.77, m
16a	2.22, s	2.21, s	2.30, s	2.31, s
17	7.41, s	7.38, s	7.57, s	7.28, s
21	5.82, s	5.82, s	5.80, s	5.85, s
24	1.13, m	1.14, m	1.15, m	1.20, overlap
	1.32, overlap	1.31, overlap	1.32, overlap	1.30, overlap
25	1.46, overlap	1.49, overlap	1.47, overlap	1.45, overlap
26	0.81 d (6.6)	0.83 d (6.6)	0.88 d (6.5)	0.88 d (6.6)
27	0.81, d (6.6)	0.79, d (6.6)	0.82, d (6.5)	0.83, d (6.6)

**Table 3 marinedrugs-17-00219-t003:** Cytotoxic activities of compounds **1**–**6**.

Compound	IC_50_ (μM)			
	SGC-7901	K562	Hela	A549
1	2.8	6.6	9.6	14.9
2	9.8	9.0	>50	24.7
3	4.7	7.6	14.1	20.9
4	20.9	16.3	29.5	33.2
5	>50	>50	>50	>50
6	>50	>50	>50	>50
Imatinib	86.8	0.2	18.8	45.6
Adriamycin	6.9	10.7	11.4	5.5

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
