# Peer review of "Divergolides T–W with Apoptosis-Inducing Activity from the Mangrove-Derived Actinomycete Streptomyces sp. KFD18"

_marinedrugs, 2019, doi:10.3390/md17040219_

Round 1
Reviewer 1 Report
1. Line 44- Authors reported structure 5 is Divergolide E and structure 6 is Divergolide A. There is a confusion about the structural representation of these two compounds. Please verify the following references, both of these references showing different structures for Divergolide E and A. Please verify it whether you put right structure of it. a.https://doi.org/10.1002/anie.201810336; b. SYNLETT 2012, 23, 2845–2849.
2. Line 61- In all the COSY spectrums of compounds, label the peaks with proton numbers of which ever protons you want to be highlighted.
3. Line 68- E configuration was assigned to ∆3 double bond for compounds 1, 2 & 3. Whereas in compound 4, ∆3 was assigned as Z. As per the data mentioned in table 2, H-3 proton coupling constants are almost similar for all compounds. Moreover, in compounds 1,2,3 if ∆3 has E- configuration both H-4a, H-3 should be spatially far off to each other and should n’t be having any correlations in ROESY spectrum. But in figure-3, ROESY correlations were observed between these two protons. That means are they spatially closure to each other (cis)? Please explain it more clearly. Even from the crystal structure of Compound 1, This double bond might have Z-configuration.
4. How does H-2, H-24 ROESY interactions decide the syn orientation of H-2 and OH-19 in structures 3 & 4?
5. Overall the manuscript needs some additional information about assigning stereo centers and configurations.
6. It is my recommendation that the manuscript needs revision and the issues pointed out need to be addressed before considering publication.
Author Response
Reply to Reviewer 1’ comments:
1. Line 44- Authors reported structure 5 is Divergolide E and structure 6 is Divergolide A. There is a confusion about the structural representation of these two compounds. Please verify the following references, both of these references showing different structures for Divergolide E and A. Please verify it whether you put right structure of it. a.https://doi.org/10.1002/anie.201810336; b. SYNLETT 2012, 23, 2845–2849.
R: We are sorry for our mistake and negligence, we have corrected the structure of 5. However, we found that in one reference compound 5 was named Divergolide E [1], while in another compound 6 was named Divergolide E [2]. According to the published date, the name of compound 6 which was early reported in the manuscript was revised to Divergolide E, while compound 5 was renamed according to its closely related analogue Divergolide U (Compound 2 in the manuscript) to 6,7-epi-24,25-dihydro-divergolide U.
Structure of Divergolide E in reference 1 Structure of Divergolide E in reference 2
1. Xu, Z.; Baunach, M.; Ding, L.; Peng, H.; Franke, J.; Hertweck, C. Biosynthetic code for divergolide assembly in a bacterial mangrove endophyte. ChemBioChem, 2014, 15, 1274-1279.
2. Zhao, G.; Li, S.; Guo, Z.; Sun, M.; Lu, C. Overexpression of div 8 increases the production and diversity of divergolides in Streptomyces sp. W112. RSC Advances. 2015, 5, 98209-98214.
2. Line 61- In all the COSY spectrums of compounds, label the peaks with proton numbers of which ever protons you want to be highlighted.
R: We have label the peaks with proton numbers in the COSY spectrums in the supporting information as suggested.
3. Line 68- E configuration was assigned to ∆3 double bond for compounds 1, 2 & 3. Whereas in compound 4, ∆3 was assigned as Z. As per the data mentioned in table 2, H-3 proton coupling constants are almost similar for all compounds. Moreover, in compounds 1,2,3 if ∆3 has E- configuration both H-4a, H-3 should be spatially far off to each other and should n’t be having any correlations in ROESY spectrum. But in figure-3, ROESY correlations were observed between these two protons. That means are they spatially closure to each other (cis)? Please explain it more clearly. Even from the crystal structure of Compound 1, This double bond might have Z-configuration.
R: We are sorry for our mistake. ∆3 double bond for compounds 1, 2 & 3 were revised to Z-configuration, Whereas in compound 4, ∆3 was revised to E.
4. How does H-2, H-24 ROESY interactions decide the syn orientation of H-2 and OH-19 in structures 3 & 4?
R: We have revised our related description as suggested.
5. Overall the manuscript needs some additional information about assigning stereo centers and configurations.
6. It is my recommendation that the manuscript needs revision and the issues pointed out need to be addressed before considering publication.

Reviewer 2 Report
The article is very well written, easy to understand.
English is excellent. I would suggest that english style of the following sentence needs some improvement (lines 57-60):
except that the Δ24 double bond of divergolide E was hydrogenated in 1, as deduced by that one olefinic quaternary carbon and one olefinic methine that corresponding to a tri-substituted double bond in divergolide E were replaced by ...
line 133: with TMS as an internal standard.
No TMS signal is visible at 0 ppm in 1H-NMR spectra included in the supplementary information. Do authors use methanol signal as secondary shift reference?. Please correct sentence accordingly.
Please add information about concentrations used for NMR analysis.
Author Response
Reply to Reviewer 2’ comments:
The article is very well written, easy to understand.
English is excellent. I would suggest that english style of the following sentence needs some improvement (lines 57-60):
1. except that the Δ24 double bond of divergolide E was hydrogenated in 1, as deduced by that one olefinic quaternary carbon and one olefinic methine that corresponding to a tri-substituted double bond in divergolide E were replaced by ...
R: Thank you for your suggestion. The sentence has been revised as suggested.
2. line 133: with TMS as an internal standard.
No TMS signal is visible at 0 ppm in 1H-NMR spectra included in the supplementary information. Do authors use methanol signal as secondary shift reference?. Please correct sentence accordingly.
R: Sorry for our mistake. Methanol signal has been used as chemical shift reference. We have correct the sentence as suggested.
3. Please add information about concentrations used for NMR analysis.
R: The information about concentrations of compounds used for NMR analysis has been added.
Reviewer 3 Report
In this manuscript, Zhao, Luo, and co-workers described isolation, structure-elucidation, and antitumor activity of divergolides T-W. Although the structures of divergolides T-W are similar to those of already isolated divergolides, the potent antitumor activity and the apoptosis-inducing mode-of-action were first reported in this manuscript. Since these important biological activity would attract readers' attention, this reviewer would like to recommend the editor to accept this manuscript after addressing the following minor points.
P 2, L 52 (and more) hydroxyl --> hydroxy. (Hydroxyl represents a radical.)
P 2, L 67: Why did the authors use ROESY instead of NOESY?
P 3, Figure 3: Is it reasonable to observe ROESY correlations between H8/H24 and H2/H24? Even from an ORTEP diagram of 1 (Figure 4), H8 looks to locate far from H24. Are there any other reasons to support the absolute configuration of C2 is R?
P 3, L 98: correlations of H4a/H3 of 3, while H4a/H2 of 4 were observed, revealing the Z and E configuration of ⊿3 double bond in 3 and 4, respectively. Insert a figure to explain ROESY correlations of 3 and 4, just like Figure 3.
P 5, L 108: Compounds 5 and 6 were inactive. Why does the Δ24 double bond make the molecule biologically inactive? Is the conformation of 1 very different from that of 5? Can the authors suggest the more detailed structural reason of inactivity of 5?
Author Response
Reply to Reviewer 3’ comments:
In this manuscript, Zhao, Luo, and co-workers described isolation, structure-elucidation, and antitumor activity of divergolides T-W. Although the structures of divergolides T-W are similar to those of already isolated divergolides, the potent antitumor activity and the apoptosis-inducing mode-of-action were first reported in this manuscript. Since these important biological activity would attract readers' attention, this reviewer would like to recommend the editor to accept this manuscript after addressing the following minor points.
1. P 2, L 52 (and more) hydroxyl --> hydroxy. (Hydroxyl represents a radical.)
R: “hydroxyl” has been revised to “hydroxy” as suggested.
2. P 2, L 67: Why did the authors use ROESY instead of NOESY?
R: In our department, ROESY instead of NOESY spectra were selected to all the compounds by NMR administrator.
3. P 3, Figure 3: Is it reasonable to observe ROESY correlations between H8/H24 and H2/H24? Even from an ORTEP diagram of 1 (Figure 4), H8 looks to locate far from H24. Are there any other reasons to support the absolute configuration of C2 is R?
R: In the manuscript, the ORTEP diagram of 1 was adjusted to a suitable direction in order to show all atoms. In fact, the distance between H8/H24 and H2/H24 are 2.886 Å and 3.170 Å respectively, which is enough to generate ROE effect.
4. P 3, L 98: correlations of H4a/H3 of 3, while H4a/H2 of 4 were observed, revealing the Z and E configuration of ⊿3 double bond in 3 and 4, respectively. Insert a figure to explain ROESY correlations of 3 and 4, just like Figure 3.
R: The figure to explain ROESY correlations of 3 and 4 was inserted as suggested.
5. P 5, L 108: Compounds 5 and 6 were inactive. Why does the Δ24 double bond make the molecule biologically inactive? Is the conformation of 1 very different from that of 5? Can the authors suggest the more detailed structural reason of inactivity of 5?
R: We are sorry for our mistake and negligence. The structure of compound 5 has been revised according to reviewer 1’ suggestion. And the configuration of C-6 and 7 in compound 5 was also different from compounds 1-4, except for the presence of Δ24 double bond. Thus, maybe the configuration of C-6 and 7 is essential for the bioactivity.
